# Influence of Trace Elements on Neurodegenerative Diseases of The Eye—The Glaucoma Model

**DOI:** 10.3390/ijms22094323

**Published:** 2021-04-21

**Authors:** Agnieszka Kamińska, Giovanni Luca Romano, Robert Rejdak, Sandrine Zweifel, Michal Fiedorowicz, Magdalena Rejdak, Anahita Bajka, Rosario Amato, Claudio Bucolo, Teresio Avitabile, Filippo Drago, Mario Damiano Toro

**Affiliations:** 1Faculty of Medical Sciences, Collegium Medicum, Cardinal Stefan Wyszyński University, 01-815 Warsaw, Poland; agnieszka.kaminska73@wp.pl (A.K.); toro.mario@email.it (M.D.T.); 2Department of Biomedical and Biotechnological Sciences, University of Catania, 95123 Catania, Italy; claudio.bucolo@unict.it (C.B.); f.drago@unict.it (F.D.); 3Center for Research in Ocular Pharmacology-CERFO, University of Catania, 95123 Catania, Italy; 4Chair and Department of General and Pediatric Ophthalmology, Medical University of Lublin, 20-079 Lublin, Poland; robertrejdak@yahoo.com; 5Department of Ophthalmology, University of Zurich, 8091 Zurich, Switzerland; sandrine.zweifel@usz.ch (S.Z.); anahita.bajka@usz.ch (A.B.); 6Mossakowski Medical Research Institute, Polish Academy of Sciences, 02-106 Warsaw, Poland; mfiedorowicz@imdik.pan.pl; 7Medical University of Warsaw, 02-106 Warsaw, Poland; rejdakmagdalena@gmail.com; 8Department of Biology, University of Pisa, 56126 Pisa, Italy; rosario.amato@biologia.unipi.it; 9Department of Ophthalmology, University of Catania, 95123 Catania, Italy; t.avitabile@unict.it

**Keywords:** glaucoma, neurodegenerative disease, iron, copper, zinc, selenium, calcium, magnesium, molybdenum, sodium, potassium, manganese

## Abstract

Glaucoma is a heterogeneous group of chronic neurodegenerative disorders characterized by a relatively selective, progressive damage to the retinal ganglion cells (RGCs) and their axons, which leads to axon loss and visual field alterations. To date, many studies have shown the role of various elements, mainly metals, in maintaining the balance of prooxidative and antioxidative processes, regulation of fluid and ion flow through cell membranes of the ocular tissues. Based on the earlier and current research results, their relationship with the development and progression of glaucoma seems obvious and is increasingly appreciated. In this review, we aimed to summarize the current evidence on the role of trace elements in the pathogenesis and prevention of glaucomatous diseases. Special attention is also paid to the genetic background associated with glaucoma-related abnormalities of physiological processes that regulate or involve the ions of elements considered as trace elements necessary for the functioning of the cells.

## 1. Introduction

Glaucoma is a heterogeneous group of chronic neurodegenerative disorders characterized by a relatively selective, progressive damage to the retinal ganglion cells (RGCs) and their axons, which leads to axon loss and visual field alterations [1,2,3,4,5,6]. It is the most common cause of irreversible blindness worldwide, affecting almost 80 million people or more than 1% of the global population [7,8,9]. It is estimated that by 2010, 1 out of 15 blind people was blind due to glaucoma, and 1 out of 45 visually impaired people was visually impaired due to glaucoma, highlighting the increasing global burden of glaucoma [4,10]. The management of glaucoma focuses on lowering IOP, the main glaucoma risk factor [5,7,11,12].

Indeed, the effectiveness of IOP-lowering therapies is questioned as they only slow down the optic nerve degeneration without significantly reversing or stopping the disease [5,7]. Thus, not only elevated IOPs play a role in the pathogenesis of glaucomatous optic nerve damage [13], which is more complex than that, and includes the contribution of many other risk factors, such as age, race, family history, environmental factors, diabetes, and myopia [1]. Moreover, many studies have shown that glaucoma is an age-related optic neuropathy showing similarities with other age-related neurodegenerations such as Alzheimer’s and Parkinson’s diseases [14]. Therefore, recent studies have also investigated the causative roles of other processes, including glutamate toxicity and glial overactivation [6,15,16,17]. Vascular dysregulation in ocular blood flow and oxidative stress have also been suggested as significant risk factors for RGCs loss in glaucoma. Moreover, mitochondrial dysfunction is another widely studied causal process in the development of glaucoma and has also been investigated as a potential drug target [9,18,19].

Another potential mechanism in glaucoma pathogenesis is alteration in the levels of various elements, mainly metals, in maintaining the balance of prooxidative and antioxidative processes, regulation of fluid and ion flow through cellular membranes of the ocular tissues [20]. Essential trace elements such as zinc, copper, selenium, manganese, chromium, cobalt, and molybdenum function as cofactors or are located in prosthetic groups of many enzymes. Metal ions such as zinc, iron, manganese, and copper are critical cofactors needed for neurotransmitter synthesis; calcium is essential for neurotransmitter release and neuroplasticity; zinc and magnesium modulate synaptic activity [16,17,18,19].

In this article, we aimed to summarize the current evidence on the role of trace elements in the pathogenesis and prevention of glaucomatous diseases. Special attention is paid to the genetic background associated with glaucoma-related abnormalities of physiological processes that regulate or involve the trace elements.

## 2. Trace Elements in Glaucoma

Numerous studies have reported altered serum and aqueous humor levels of trace elements in glaucoma patients (Table 1).

Both increased and decreased concentrations had been observed by Lee et al. [43], Ceylan et al. [69], and Akyol et al. [70]. Ceylan et al. aimed to determine whether trace element levels have a role in the development of pseudoexfoliation (PEX) syndrome and pseudoexfoliation glaucoma (PEG) [69]. Levels of zinc, copper, selenium, manganese, chromium, cobalt, molybdenum, nickel, vanadium, arsenic, aluminum, mercury, cadmium, and strontium were determined in serum samples of PEX and PEG patients and control subjects using inductively coupled plasma–mass spectrometry. Mn, Mo, and Hg concentrations were found to be significantly elevated in patients with PEX. According to the authors, this may suggest the role of manganese and mercury as the “strongest determinants of PEX and molybdenum as the strongest determinant of PEG”. However, it should be noted that it is not clear what the term “determinants” would mean. No precise role of these elements in the mechanism of glaucoma formation has yet been proven. Also, Hohberger et al. analyzed concentrations of trace elements in aqueous humor samples of patients with primary open-angle glaucoma (POAG) and PEG [20]. The levels of cadmium, iron, manganese, cobalt, copper and zinc were measured by Plasma-Mass-Spectrometry. Patients with POAG and PEG had significantly higher aqueous humor levels of zinc compared to healthy controls. Iron level was significantly reduced in PEG and a significant difference between POAG and PEG was observed. According to the authors, the differences detected support the hypothesis that these elements are involved in the pathogenesis of open-angle glaucoma. No significant differences were observed in aqueous humor levels of manganese, cobalt, copper and cadmium between glaucoma patients and controls.

Vinetskaia and Iomdina used the method of spectral analysis to measure the content and composition of micro- and macro-elements in samples of lacrimal fluid from adults with open-angle glaucoma and diabetic retinopathy [71]. Changed iron, magnesium, and aluminum levels were detected and, to a lesser extent, zinc and copper in patients with diabetic retinopathy. In contrast, open-angle glaucoma was associated with reduced zinc and increased iron content.

The studies on a pigmentary dispersion animal model of PEG (DBA/2J mice) provided the basis for comparison between healthy and glaucomatous subjects in humans (Figure 1).

DeToma et al. measured the concentrations of iron, copper, zinc, magnesium, manganese, and calcium in the retina and in the retinal projection of pre-glaucomatous and glaucomatous DBA/2J mice and age-matched controls using inductively coupled plasma mass spectrometry (ICPMS) [36]. Glaucomatous mice had lower retinal Fe concentration than pre-glaucomatous and control mice. Retinas of pre-glaucomatous mice had greater Mg, Ca, and Zn concentrations than those of glaucomatous and greater Mg and Ca than controls. Retinal Mn levels were significantly lower in glaucomatous mice compared to controls and pre-glaucomatous mice. The superior colliculus (SC, the primary retinal target in rodents and one of the earliest sites of pathology in the DBA/2J mouse) of the control mice contained more Fe, Mg, Mn, and Zn than that SC of glaucoma model DBA/2J mice.

### 2.1. Iron

Iron (Fe) is an essential element found in the prosthetic groups of many important metalloproteins: hemoglobin, myoglobin, and the active centers of numerous enzymes such as catalase, peroxidases, and cytochromes. Iron acts as a target for both nitrogen monoxide and carbon monoxide, two messenger molecules with crucial roles in a variety of conditions, including glaucoma [72]. Iron also participates in processes related to neurodegeneration and neuroprotection [45,73].

Iron is a redox-active agent. On the contrary, copper and zinc support the reduction of free radicals via superoxide dismutase [44,74]. The excessive consumption of iron may increase the risk of glaucoma. Thaler et al. showed that rats receiving iron before a partial optic nerve crush had significantly greater damage of the RGCs than the control group [75]. On the other hand, the iron level was significantly reduced in PEXG patients and a difference was seen when comparing POAG and PEXG individuals. The possible mechanisms of iron’s effect in glaucoma may include the activity of enzymes and other proteins involved in iron metabolism. Lin et al. (2014) investigated the association between serum ferritin level and the likelihood of a glaucoma diagnosis [76]. Their study showed that a higher serum ferritin level was associated with a greater frequency of glaucoma in a representative sample of South Koreans. The participants in this population-based study with higher serum ferritin levels (greater than 61 ng/mL) had significantly higher incidence of glaucoma than those with lower levels (less than 31 ng/mL). As the authors claimed, this finding may “help to elucidate the pathogenesis and lead to novel therapeutic approaches for glaucomatous disease”. Gye et al. investigated the association between serum ferritin levels and glaucoma in a South Korean population [22]. The studies also considered other factors (age, white blood cell count, C-reactive protein, and total vitamin D), including those related to iron metabolism (serum iron, total iron-binding capacity, transferrin saturation). It was shown that males with high serum ferritin had a higher risk for glaucoma than those with low ferritin; no difference was observed for women. Other markers of iron metabolism, such as iron level, transferrin saturation, and inflammation measures, were not associated with glaucoma. They concluded that a high serum ferritin level was associated with a high risk of glaucoma. This effect might have been related to oxidative stress and inflammation, which play a role in glaucoma development.

### 2.2. Copper

Copper (Cu) is generally considered to have antioxidant roles in human metabolism, as this metal is involved in eliminating superoxide radicals by the superoxide dismutase (SOD) enzyme. Cu is built-in in the active center of such enzymes as: cytochrome c oxidase; β-dopamine hydroxylase; superoxide dismutase (SOD1, SOD3); ceruloplasmin and hephaestin; tyrosinase; and lysyl oxidase [44,77]. Copper takes part in Fenton’s reaction, which forms the highly reactive hydroxyl radical. This reaction is well known for involving reduced iron as the major element that catalyzes the break-down of hydrogen peroxide resulting in the active production of free radicals [78]. Proteins that are involved in cellular copper metabolism include, among others, membrane transporters CTR1 and CTR2, metallo-chaperons, and metallothionein.

Normal copper metabolism is essential to ocular tissue health and changes in pigmentary retinopathies and high myopia [64]. Silverstone et al. measured the blood concentrations of zinc and copper and the urine concentration of copper in different groups of highly myopic patients, and in highly myopic patients with retinal detachment, showing high levels of zinc and low levels of copper in serum [65]. In the high myopic group with retinal detachment, serum zinc and copper concentrations were significantly elevated. The possible role of Cu in the development of PEX was investigated by Yildirim et al. [79], who measured the concentration of zinc and copper in lenses removed from the cataract patients. However, in contrast to Zn, the concentration in lenses from patients with PSX was significantly lower than that in patients without PSX; the mean concentration of copper did not differ significantly. Serum and aqueous humor zinc and copper concentrations of patients with glaucoma and cataract were determined by Akyol et al. [70]. The highest mean copper concentration was found in the glaucoma group. In addition, there was a significant negative correlation between the aqueous humor levels of zinc and copper in patients with glaucoma. It was concluded that an increased copper concentration and low zinc concentration might be of importance in patients with glaucoma. Since copper deficiencies have been linked with photoreceptor loss and myopia with increased scleral wall elasticity, and differences have been previously shown in Fe content in glaucomatous and healthy patients, DeToma et al. measured the levels of iron (Fe) and copper (Cu) in ocular structures of healthy [36], pre-glaucomatous and glaucomatous DBA/2J mice. No differences in retinal Cu levels were shown between any of the groups in the study but the glaucomatous mice had lower retinal Fe concentrations than pre-glaucomatous and control mice.

The concentration of copper and iron was also measured in the aqueous humor of rabbit eyes treated with steroids; the Cu and Fe levels in relation to a steroid-induced increase in intraocular pressure (IOP) were evaluated [29]. After 30 days of steroid treatment, the mean concentration of copper in the aqueous humor of steroid-treated rabbits was significantly lower compared to the control group. However, the concentration of iron was not different. As concluded, lowering copper concentration resulting from steroid treatment may play an important role in the maintenance of high IOP in rabbits’ eyes, which may explain the role of Cu in the pathogenesis of OAG.

### 2.3. Calcium

Calcium participates in an intracellular signaling system by acting as a diffusible second messenger. Calcium and calcium signaling play a crucial role in the nervous system [80]. In neurons, intercellular calcium signals—the transient changes in cytoplasmic free calcium—propagates the calcium signal to neighboring cells. Moreover, activation of calpain, a ubiquitous calcium-sensitive protease, is known to play a role in neurodegenerative diseases such as cerebral ischemia, Alzheimer’s disease, Parkinson’s disease, Huntington’s disease, multiple sclerosis, and others [81]. There is now evidence that the pathological progression of neurodegeneration in glaucoma also includes the Ca^2+^-dependent mechanisms [14]. In contrast to some other macro- or micronutrients indispensable to life, the mechanism of calcium’s effect on the incidence of glaucoma and its course has been intensively studied [82]. The relationship between glaucoma occurrence and excessive consumption of calcium has been reported; on the other hand, calcium supplementation led to a reduction in human susceptibility to glaucoma [5,83].

Crish and Calkins reviewed the results of studies showing that the pathological neurodegeneration in glaucoma might result from Ca^2+^ dependent mechanisms [14]. Calcium homeostasis and calcium signaling are related closely to neuronal functions such as synaptic transmission or cell survival. It was shown that disturbed calcium homeostasis might make neurons more vulnerable to oxidative stress [84]. It is supposed that the calcium channel-blocking agents (e.g., nimodipine and brovincamine) or nilvadipine (an L-type Ca^2+^ blocking agent) may retard the age-related disturbance of calcium homeostasis and thus decrease the incidence of open-angle glaucoma, which is a well-known age-dependent disease.

Plasma membrane calcium channel inhibitors were found to arrest acute axonal degeneration and improve regeneration after damage to the optic nerve [50,59]. Numerous experiments have shown that the application of calcium channel blockers (CCBs, e.g., verapamil) caused significant reduction of intraocular pressure. These generally cause widening of blood vessels in the eye and increase ocular blood flow in experimental animals and patients with open-angle glaucoma. CCBs, such as nimodipine, brovincamine, and nilvadipine, had beneficial effects on visual function not only in normal humans but also in patients suffering from glaucoma. In vitro studies showed that calcium channel blockers exerted protective effects on neurons undergoing apoptosis and necrosis. In experimental animal models also, neuroprotective effects were shown. In these experiments, beta-adrenoceptor antagonists (e.g., betaxolol) show calcium channel blocking activity, which may be responsible for the neuroprotective effects (Araie and Mayama) [85]. The results obtained by Ganekal on rabbits topically treated with verapamil [86] and diltiazem showed that these calcium channel blocking agents significantly reduced intraocular pressure in steroid-induced glaucoma.

The role of mitochondrial dysfunction has been studied in the development of glaucoma. Many results indicated that reactive calcium signaling and the generation of free radicals and other active oxygen species regulated by the mitochondria [28,81] are critical in glaucoma pathogenesis [9]. The authors suggest that extensive studies of glaucoma and its relationship with aging, mitochondrial diseases, and calcium signaling dysfunction should lead to the development of new drugs based on calcium channel regulators used to treat glaucoma and other neurodegenerative diseases of the retina. They suggest that the utilization of mitochondrial-specific calcium channel regulators may help treat glaucoma and other neurodegenerative eye diseases.

Calcium entry blockers are probably the most frequently and most accurately tested substances used as anti-glaucoma medications. Ca^2+^ entry blockers are potent drugs dilating ocular blood vessels, which have been demonstrated both in in vivo and in vitro studies [85,87]. This process is called vasodilation relaxation of the smooth muscles in ocular vessels and depends mainly on the extracellular Ca^2+^ flow. Ca^2+^ entry blockers may play a role in relaxing the retinal, long posterior ciliary arteries to improve the ocular circulation in vascular diseases manifested by increased tension of the eye blood vessels. Treatment with nimodipine, a centrally active calcium channel antagonist, significantly improved the visual field and color vision in glaucoma patients [88]. However, this drug was not shown to alter the blood flow dynamics in macular vessels. Yamamoto et al. investigated the effect of nilvadipine [89], a calcium-channel blocker, on the hemodynamics of retrobulbar vessels in normal-tension glaucoma. Oral nilvadipine increased the blood flow in distal retrobulbar arteries in normal-tension glaucoma without affecting more proximal blood vessels. The conclusion was that nilvadipine might have a beneficial effect on the blood flow in retrobulbar vessels in normal-tension glaucoma.

In the experiments carried out both in vitro and in vivo on animal models, Hara et al. reported that lomerizine [23,31], a Ca^2+^ channel blocker, increased ocular circulation and protected neuronal cells. In rabbits with disturbed circulation, lomerizine and other Ca^2+^ channel blockers increased the ocular circulation and protected the optic nerves. Moreover, lomerizine reduced damage of the retina in rats, presumably through a Ca^2+^ channel blocking effect, which may involve an improvement in the ocular circulation. The authors claim that lomerizine may be useful as a therapeutic drug against ischemic retinal diseases (such as glaucoma and retinal vascular occlusive diseases) that involve a disturbance of the ocular circulation. Kitazawa et al. evaluated the effects of nifedipine, a Ca^2+^ antagonist, on the visual field in low-tension glaucoma [32]. Patients received 30 mg/day per os for 6 months. Six patients showed a constant improvement in the visual field. Of the substances known to benefit the survival of damaged neurons used by Osborne et al. to treat glaucoma [90], beta-blockers were shown to reduce the sodium and calcium influx process. Betaxolol was shown to be the most effective antiglaucoma medicine, reducing sodium and calcium influx. Topical application to rat eyes of levobetaxolol and timolol, beta-adrenergic receptor inhibitors (beta-blockers), both known to influence sodium and calcium flux in cell membranes, was shown to alleviate the effects of ischemia-reperfusion injury. Based on these results, the authors concluded that these drugs can attenuate ganglion cell death in glaucoma. They also suggest that betaxolol and, to a lesser extent, timolol may be effective neuroprotectants and that the mechanism of their action is by blocking the calcium and sodium influx into neural cells. Studies carried out by Osborne et al. on the isolated optic nerve also showed that beta-blockers which reduce the influx of sodium are particularly effective neuroprotectants [90]. They also reduce the influx of calcium which additionally benefits the survival of insulted neurons. Of the used antiglaucoma drugs, betaxolol was shown to be the most effective substance at reducing sodium/calcium influx.

Siegner et al. determined the effect of different calcium channel blockers (representative members from six classes) on intraocular pressure in the eyes of the macaque (*Macaca fascicularis*) [63]. Moreover, the effect of other antiglaucoma medications was studied in combination with verapamil to determine their effect on intraocular pressure. This study showed that all classes of calcium channel blockers significantly lowered intraocular pressure. The conclusion was that calcium channel blockers and combinations of antiglaucoma medications with verapamil might provide a useful medication for reducing intraocular pressure in patients with ocular hypertension or POAG. In monkeys also, but with laser-induced unilateral glaucoma, flunarizine, a nonselective calcium channel blocker, was shown to reduce intraocular pressure in a dose-dependent manner when administered to glaucomatous eyes [91]. However, in this case, a hypotensive effect on the normal, untreated eyes was also observed.

Different experimental approaches have been used to investigate the mechanism of the interaction of drugs used to treat glaucoma on the functioning of calcium channels in nerve cells. Melena et al. investigated the affinity of several antiglaucoma drugs, including betaxolol, timolol, and others, for voltage-dependent Ca^2+^ channels (VDCCs) using radioligand binding assays [92]. Of the antiglaucoma drugs investigated, betaxolol displayed the greatest L-type VDCC-blocking activity, and this may account for some of its actions in ocular hemodynamics. Hong et al. investigated various commercial antiglaucoma drugs and their effects in inhibiting glutamate-induced intracellular free Ca^2+^ increase in cultured N1E-115 neuroblastoma cells [93]. They found that betaxolol, dipivefrin, brimonidine and timolol have remarkable effects on inhibiting the glutamate-induced Ca^2+^ increase, and also decreased the basal calcium concentration in the neuroblastoma cells in vitro. These results indicate that betaxolol, dipivefrin, and brimonidine may have neuroprotective effects in inhibiting the glutamate-induced cell damage resulting from the excessive Ca^2+^ influx.

The effect of calcium channel blockers on the visual field, optic discs, and intraocular pressure was analyzed by Liu et al. [94]. There was no apparent difference in intraocular pressure, visual field, or optic discs when comparing patients using calcium channel blockers with the control group. Thus, this study showed no beneficial effect of calcium channel blockers on the course of the disease in patients suffering from glaucoma. The results of most of these studies showed that calcium channel blockers are effective in glaucoma treatment. They have shown a positive effect on ocular blood flow and visual field in normotensive glaucoma (NTG) [13,95]. However, some unwanted side effects such as hypotension and bradycardia restricted the use of calcium channel blockers for the treatment of glaucoma [13,96].

### 2.4. Magnesium

Magnesium plays a significant role as a cofactor for more than 350 enzymes [97]. The enzymes involved in ATP production and hydrolysis are also magnesium-dependent. Magnesium deficiency also results in increased oxidative stress and inducible NOS stimulation that can further contribute to the initiation and progression of ocular pathologies such as cataract, glaucoma, and diabetic retinopathy. Moreover, magnesium deficiency may be the cause of vasoconstriction. This can lead to tissue ischemia and tissue death, which is one of the pathogenic factors in POAG. Furthermore, magnesium deficiency is a causative factor in increasing oxidative stress and inducing nitric oxide synthase (NOS) activity that can further contribute to the initiation and progression of ocular pathologies such as cataract, glaucoma, and diabetic retinopathy. Reduced Na^+^/K^+^-ATPase activity associated with magnesium deficiency has been shown to cause defective neurotransmitter transport mechanism, mitochondrial dysfunction, defective Golgi body function, protein processing dysfunction, neuronal degeneration, and apoptosis [41]. Moreover, the loss of Na^+^/K^+^-ATPase, resulting from Mg deficiency, causes intracellular Na^+^ accumulation and release of mitochondrial Ca^2+^ ions, which results in increased cytoplasmic Na^+^ and Ca^2+^ [13,98]. Increased intracellular Ca^2+^ and Na^+^ further cause cellular swelling leading to RGCs apoptosis.

The improvement of ocular blood flow and prevention of ganglion cell loss make magnesium a good candidate for glaucoma treatment [13]. Mg improves blood flow by modifying endothelial function via endothelin-1 (ET-1) and endothelial nitric oxide (NO) pathways. Mg also exhibits a neuroprotective role by blocking N-methyl-D-aspartate (NMDA) receptor-related calcium influx and inhibiting glutamate release. It protects the cell against oxidative stress and apoptosis.

Mg is important for maintaining the structural and functional integrity of several vital ocular tissues [13,97]. The magnesium content in the lens, especially in its peripheral part, is higher than that in the aqueous and vitreous humor of the eye. Magnesium has also been shown to play a critically important role in retinal functions. Reduction in blood flow was observed in ocular tissues—retina, optic nerve, iris, and choroid—in various ocular disorders, including glaucoma. Mg was shown to enhance ocular vasodilation and optimize ocular blood flow. Therefore, Mg supplementation may have a therapeutic value in glaucoma and may protect the ocular tissues by regulating ocular blood flow and reducing oxidative stress. This is supported by the finding that magnesium facilitates blood flow by reducing cytokine levels and free radical production and preventing intracellular calcium entry [99].

Head reported that glaucoma patients for whom the optic nerve damage is caused by the decreased blood supply to the optic nerve could benefit from supplemental magnesium, which is a “physiological calcium blocker”. Gaspar et al. evaluated the effect of magnesium in POAG patients and those with normal-tension glaucoma [100]. Administration of magnesium twice a day for one month had beneficial effects on the visual field. Moreover, all parameters studied, blood cell velocity, cold-induced blood flow cessation, and the number of capillaries per microscopic field and digital temperature, improved significantly. The authors concluded that magnesium improves the peripheral circulation, widens the visual field, and has a relieving effect on glaucoma patients.

Summing up, the above data speaks for the beneficial effect of magnesium in relieving glaucoma symptoms. Mg may protect RGCs from oxidative injury by combined effects on voltage-dependent calcium channels, glutathione synthesis, lipid peroxidation, and by maintaining the regulation of many metabolic enzymatic reactions. Both improvements in ocular blood flow and prevention of ganglion cell loss would make magnesium a good candidate for glaucoma treatment. However, as most authors emphasize, further studies are needed on the possible beneficial effect of Mg in curing glaucoma. What is more, some disappointing results have been obtained. In the studies involving normal-tension glaucoma (NTG) patients, Aydin et al. have shown that treatment with 300 mg of magnesium citrate for one month did not change the ocular blood flow (although it caused some improvement in the visual field) [101]. These results showed that oral magnesium therapy might improve the visual field in normotensive glaucoma, but it does not seem to affect ocular blood flow in patients with NTG. Another mechanism could therefore contribute to this effect.

### 2.5. Molybdenum

The concentrations of molybdenum (Mo) were significantly increased in the blood of patients with PEX [69]. Based on this result, the authors postulated molybdenum as a strong determinant of PEX. Molybdenum is part of some of the metallo-flavoproteins—xanthine oxidase, involved in purine metabolism, and (besides iron) aldehyde dehydrogenase, catalyzing the oxidation of acetaldehyde to acetic acid. However, so far, no precise role of Mo in glaucoma pathophysiology has been proven.

### 2.6. Zinc

Zinc (Zn) is an essential microelement present in the active centers of about 200 enzymes, involved in many signaling pathways [102]. Zinc is generally considered an antioxidant factor involved in the process of elimination of superoxide radicals [103]. Zinc is a trace element that appears to play an integral role in maintaining normal ocular function [37]. In the eye, zinc is essential in the retina, choroid, cornea, and lens [104]. In the retina and retinal pigment epithelium, zinc interacts with taurine and vitamin A and modifies photoreceptor plasma membranes, regulates the light-rhodopsin reaction, modulates synaptic transmission, and serves as an antioxidant. Because of its elevated levels in POAG and PEG, Zn may be related to glaucoma pathogenesis. Matrix metalloproteinases are zinc-containing enzymes capable of degrading all kinds of extracellular matrix proteins and processing several bioactive protein molecules, some of which play a decisive role in glaucoma pathogenesis.

Newsome et al. and Prasad demonstrated that zinc concentrations are reduced in human eyes with signs of age-related macular degeneration (AMD) and suggested that zinc deficiency may lead to oxidative stress and retinal damage [34,38]. Studies of dietary supplementation of zinc gave ambiguous results, but it seems that it can help alleviate the effects of AMD in the elderly.

As reviewed by Brown et al. [33], zinc has a role in retinal metabolism and may be beneficial in macular degeneration. Research cited in this review showed that altered serum levels of some trace elements, e.g., copper and zinc, may impact the incidence and progression of glaucoma disease. Both increased and decreased concentrations have been observed to affect the illness [47,73,74].

Zinc seems to play a role in glaucoma pathophysiology [105]. In some forms of glaucoma, the cones of human eyes were shown to contain high amounts of zinc, and the damage to blue cones might cause the release of this metal to the extracellular space [58]. Zinc could also be involved in light-induced retinal injury; however, the mechanisms of retinal light damage in the pathology of glaucoma remain unknown [106]. Neurons overloaded with zinc probably exist in the mammalian retina; however, it appears unlikely that the death of RGCs after ischemia, which occurs in glaucoma, is due to an influx of zinc originating from zinc-enriched neurons. The mean concentration of zinc in the lens from patients with PEX was significantly lower than that in the lenses of patients without PEX. The decreased content of zinc could increase oxidative stress that is thought to contribute to the development of PEX in cataract patients [79].

### 2.7. Selenium

Selenium (Se) is an essential mineral incorporated into at least 25 distinct proteins, mainly as selenocysteine residue. These proteins include glutathione peroxidases (GPx), iodothyronine deiodinases, and thioredoxin reductases. Se is best known for its crucial role in the GPx enzyme system, the major antioxidant defense system. Selenium is also incorporated into other selenoproteins, which prevent cellular damage through their antioxidative properties [49,107].

It was found that high plasma Se is associated with glaucoma and the intake of selenium might increase the risk of glaucoma [49]. Selenium supplementation was shown to be linked to glaucoma [46,47]. Moreover, selenium concentration in aqueous humor of patients with POAG and PEG was higher than in healthy individuals. Bruhn et al. compared selenium levels in blood plasma and aqueous humor in patients with and without POAG [108]. High plasma selenium concentration was significantly associated with glaucoma. Yilmaz et al. investigated the levels of selenium in aqueous humor, conjunctiva, and serum of patients with PEX and control subjects [109]. The mean Se levels in aqueous humor of patients with PEX syndrome were significantly lower than in the control group. However, serum Se levels in PEX patients and healthy controls did not differ significantly (a tendency toward lower Se in PEX was observed). The authors conclude that the reduced levels of Se may suggest the impairment of the antioxidant defense system in the PEX.

Laganovska et al. estimated the serum kynurenine, neopterin, and selenium concentrations in aqueous humor from the anterior chamber of the eye, and selenium content in lenses from operated cataract patients with and without PEX [42]. Significantly increased kynurenine and neopterin concentrations were observed in patients with PES compared to healthy controls. They also had the lowest content of selenium in the serum and lens compared with patients without PEX. The decreased content of selenium may elicit immune system activation via increased oxidative stress, as was indicated by the increased concentration of kynurenine and neopterin.

Conley et al. determined the effects of selenium on trabecular meshwork cells, a likely site of pathology for glaucoma [110]. Human trabecular meshwork (HTM) cells were treated with selenium, in the form of methyl-seleninic acid (MSeA), at a concentration considered to be physiological. Selenium uptake by cells, alterations in protein secretion, intracellular signaling, and cell morphology were monitored. Moreover, the role of integrin signaling in MSeA-treated cells was investigated. Selenium-induced morphological changes occurred before alterations in protein secretion and intracellular signaling. Selenium was shown to influence several indicators of HTM cell homeostasis but did not affect cell viability.

The association between selenium and glaucoma seems to be complex and not well-understood [61]. In the study performed in 1996, involving 1312 patients, selenium supplementation (200 mg/daily) was linked to the development of glaucoma. The risk was even higher in those who continued selenium supplementation after the trial. Other case-control studies cited in this article (Bruhn et al.) also showed that selenium supplementation might carry a risk of glaucoma [108]. In summary, there are both biological and human studies suggesting that selenium supplementation is linked with an increased incidence of glaucoma. The association between high plasma selenium level and glaucoma confirms the suggestion that selenium supplementation may carry a risk of developing glaucoma.

### 2.8. Sodium and Potassium

Sodium ions (Na^+^) are involved in neuronal impulse transmission, affect the osmotic pressure of body fluids, and increase the hydration of cellular colloids. Potassium cations (K^+^) also participate in the transmission of impulses by neurons. Early studies showed the differences in plasma potassium concentration between normal subjects and glaucoma patients [111]. This could suggest a critical role of potassium ions in the pathogenesis of glaucoma. Indeed, numerous subsequent studies have confirmed the important role in glaucoma of this element, in cooperation with other ions (Na^+^, Mg^+,^ and Ca^2+^) and their transport through cellular and mitochondrial membranes.

Potassium (K^+^) channels play key roles in modulating the electrical properties of RG cells [112]. A number of eye diseases, including glaucoma and ischemic optic neuropathy, may cause injury or the death of these cells, and subsequently permanent visual dysfunction. Studies have identified a number of K^+^ channels in RG cells, including inwardly rectifying K^+^ (Kir), ATP sensitive K^+^ (K_ATP_), tandem pore domain K^+^ (TASK), voltage-gated K^+^ (Kv), etheràgogo (Eag), and Ca^2+^ activated K^+^ (K/Ca) channels, which are involved in the controlling of RGCs excitability, their maintenance, survival, and neuroprotection. Considering these important roles of K^+^ channels, it is supposed that the study of their regulation in RG cells may result in explaining some pathophysiological mechanisms and provide new means of their protection in glaucoma patients.

In early studies, O’Donnell et al. conducted experiments to evaluate the presence of Na/K/Cl co-transport activity in trabecular meshwork (TM) cells and to test the hypothesis that modulation of Na/K co-transport alters intracellular volume and, consequently, permeability of the TM cell membranes [113]. They demonstrated that bovine and human TM cells exhibit Na/K/Cl co-transport activity that is modulated by a variety of hormones and neurotransmitters. These findings suggest that Na/K/Cl co-transport of TM cells is of central importance to the regulation of intracellular volume and TM permeability. Defects of Na/K/Cl cotransport may underlie the pathophysiology of glaucoma.

Magnesium plays a crucial role in Na^+^ and K^+^ transport in cells. It is an important cofactor of Na^+^/K^+^-ATPase, responsible for the active transport of Na^+^ out of the cell membrane in exchange with K^+^ [13,114]. Mg^2+^ deficiency leads to functional loss of Na^+^/K^+^-ATPase and causes intracellular Na^+^ accumulation and release of mitochondrial Ca^2+^ ions, which results in increased cytoplasmic Na^+^ and Ca^2+^ [98]. Increased intracellular Ca^2+^ and Na^+^ further cause cellular swelling leading to retinal ganglion cell apoptosis.

An excessive influx of sodium inside the cells through voltage-gated sodium channels is an important event in the cascade leading to degeneration of axons. Hains and Waxman tested the hypothesis that sodium channel blockade with phenytoin would result in neuroprotection of RGCs and optic nerve axons in an experimental model of glaucoma in Wistar rats with chronic elevation of intraocular pressure (IOP), leading to optic nerve damage [48]. The study showed that orally delivered phenytoin was effective in protecting neurons. Saito et al. investigated whether NS-7 (4-(4-fluorophenyl)-2-methyl-6-(5-piperidinopentyloxy) pyrimidine hydrochloride [55], a novel Na^+^/Ca^2+^ channel blocker, can protect the rat retina. They showed that NS-7 has neuroprotective effects against retinal damage resulting from subjection to ischemia. The authors also suggested that NS-7, can be used as an agent for treating acute ischemic retinopathy, including diseases associated with high intraocular pressure, such as acute angle-closure glaucoma.

Dong and Hare characterized the role of intracellular Na^+^ overload in the ischemic injury of acutely isolated rat optic nerves by evaluating electrically potentials (CAPs) from the optic nerves [115], a measure of optic nerve function. When a rapidly reversible Na^+^ channel blocker was present during one hour of oxygen and glucose deprivation, the recovery of the potentials was significantly enhanced. The results showed that intracellular Na^+^ overload played a significant role in the ischemic injury of optic nerves. This may depend at least partially upon Ca^2+^ influx from the extracellular space.

Studies on the isolated optic nerve showed that substances that reduce the influx of sodium were effective neuroprotectants [90]; beta-blockers were shown to reduce sodium influx into cells. They also reduce the influx of calcium which additionally benefits the survival of insulted neurons. Of the used antiglaucoma drugs, betaxolol was the most effective at reducing sodium/calcium influx. Topical application of levobetaxolol and timolol to rats, which attenuated the effects of ischemia injury, also appeared highly effective. These studies showed that levobetaxolol is a more effective neuroprotectant than timolol because of its greater capacity to block sodium and calcium influx.

K_ATP_ channel openers are considered as potential therapeutics for the treatment of glaucoma. They were shown to lower intraocular pressure (IOP) in animal models and in vitro cultured anterior segments of human eyes. A study by Chowdhury et al. showed that cromakalim is a hypotensive agent acting via activation of Kir6.2 containing K_ATP_ channels [51]. Its effect is additive in combination with the commonly used anti-glaucoma drug latanoprost. In another study, Chowdhury et al. have used derivatives of the K_ATP_ channel opener cromakalim and evaluated their intraocular pressure lowering capabilities [52]. The phosphate derivatives of cromakalim proved to be effective in lowering IOP in mouse and rabbit models in vivo. No toxic effects on cell structure or the aqueous humor outflow pathway were observed after the treatment, suggesting that these drugs are strong candidates for ocular hypotensive agents. The recent study by Chowdhury et al. showed that other K_ATP_ channel openers—diazoxide and nicorandill—lower intraocular pressure by specifically activating the Erk1/2 pathway in ocular cells [53,54], the signaling axis, which is one of the most important pathways involved in survival and proliferation of various cells. The obtained results indicate a mechanism according to which K_ATP_ channel openers, targeting the Erk1/2 signal transduction pathway, can treat hypertensive ocular diseases such as glaucoma.

Katz et al. described the Na^+^/K^+^-ATPase inhibitor, digoxin, selective for the α2β3 isoform of the enzyme [116], and showed that its application efficiently reduces the pharmacologically induced and basal intraocular pressure in rabbit eyes. This work confirms that selective digoxin derivatives, which effectively inhibit the Na^+^/K^+^-ATPase and reduce aqueous humor production in the eye, may have the potential for glaucoma drug therapy. Skatchkov et al. and Yamauchi et al. showed that ATP-dependent potassium channels (K_ATP_ channels) in the mitochondrial or plasma membranes might provide protection against retinal ischemia [66,117]. In the Müller glial cells, K_ATP_ channels regulate retinal current and play a key role in retinal protection against ischemic conditions, e.g., ischemic insult [56,66,117]. In other studies, MaxiK channels and K_ATP_ channels were found in the eye trabecular meshwork cells [118,119], and the opening of these potassium channels, it was postulated, might play a protective role by increasing the uveal outflow.

Choi et al. investigated the effect of KR-31378 ((2S,3S,4R)-N’’-cyano-N(6-amino-3,4dihydro-3-hydroxy-2-methyl-2-dimethoxymethyl-2Hbenzopyran-4-yl)-N’-benzyl-guanidine) a benzopyran analog [120] and a potent K_ATP_-channel opener, on reducing intraocular pressure and its protective effect on RGCs. They showed that under the condition of chronic ischemia, the cell density in the KR-31378–treated group was higher than that in the non-treated group, and intraocular pressure was reduced. In the acute retinal ischemia model RGCs in KR-31378–treated retina were protected from ischemic damage. These results showed that KR-31378 exerts both a neuroprotective effect and a pressure-reducing effect. The authors suggest that KR-31378 may be used to improve glaucoma therapies.

Functional disorders of K_ATP_ channels can be counted as events leading to glaucoma; these channels may also play a role in regulating intraocular pressure. K_ATP_ channel openers were shown to have both hypotensive and neuroprotective properties. Therefore, they have the potential to be a new class of glaucoma therapeutics. Roy Chowdhury et al. aimed to determine whether the opening of K_ATP_ channels affected the trabecular meshwork and intraocular pressure regulation. They treated in vitro the human anterior segment perfusates with K_ATP_ channel openers—diazoxide, nicorandil, and cromakalim. All these treatments showed a reduction in pressure (by 30–40%) and outflow facility (by 50–80%). In a similar experiment, latanoprost, the active component of the glaucoma drug Xalatan, increased the outflow facility by 67% [121]. Pressure reduction with K_ATP_ channel openers was reversible and was completely inhibited by the channel closer glibenclamide. Their results have shown that K_ATP_ channel openers have the potential to become a new class of glaucoma therapeutics that can lower intraocular pressure and protect the RGCs and the optic nerve from neuronal degeneration. Putney et al. evaluated the effects of dexamethasone treatment on Na/K/Cl co-transport activity and co-transporter protein expression in trabecular meshwork (TM) cells [122]. The authors found that the exposition of human and bovine TM cells to dexamethasone stimulates Na/K/Cl co-transport activity. Moreover, they found that two–five days of treatment with dexamethasone increased the co-transporter protein expression while longer exposures decreased the protein levels below controls’. The authors’ findings suggest that dexamethasone may be exerting its effect by altering Na/K/Cl cotransport function in TM cells, which may be an underlying factor in steroid-induced glaucoma.

### 2.9. Manganese

Manganese (Mn) is a redox-active metal and can promote free radical formation, leading to oxidative stress. Manganese is a component of the superoxide dismutases (MnSODs), which catalyze the dismutation reaction of superoxide anion radical to hydrogen peroxide and oxygen. Concentrations of manganese have a significant effect on the activity of antioxidant enzymes and thus on the fight against oxidative stress [123].

Ceylan et al. found that manganese concentration was significantly increased in patients with PEX [69]. The authors suggested that Mn levels had a strong association with PEX and that the increased levels of serum Mn may have a possible role in the pathobiology of PEX. When the three groups (PEX, PEG, and control) were analyzed together, Mn concentration was significantly different. On the other hand, they observed that Mn levels were almost the same in the PEG and the control groups. Moreover, concentrations of all elements measured fell into the respective reference intervals, except for the Mn level of PEX patients, which was slightly above the upper limit for Mn. On the other hand, as reported by Lin et al., blood manganese level was negatively associated with glaucoma diagnosis in a population-based study on 2680 South Korean individuals [124]. These findings suggest that a low blood manganese level could be associated with greater susceptibility to glaucoma. This element may have some, as yet undetermined, relationship with the occurrence of this disease.

### 2.10. Heavy Metals

In biology and medicine, the term “heavy metals” generally refers to elements used in industry and characterized by toxicity to humans or the environment. Heavy metals include metals, e.g., mercury (Hg), lead (Pb), cadmium (Cd), chromium (Cr), nickel (Ni), copper (Cu), zinc (Zn), and bismuth (Bi); semi-metals, e.g., arsenic (As), tellurium (Te); and even non-metals (selenium; Se). The toxic action of these “metals” is associated with their ability to accumulate in the body, including bones, kidneys, and brain. Their salts and oxides can cause severe poisoning and acute and chronic diseases of the circulatory system, nervous system, and kidneys.

Heavy metals have the capacity to replace previously bound metals and change the concentrations of other metals in patients’ tissues, including ocular tissues [125]. Therefore, in several publications appearing in ophthalmic literature, the influence of heavy metals on the incidence and morbidity of glaucoma was examined.

In the study by Ceylan et al., the concentration of several trace elements was measured in the blood of PEX and PEG patients [69]. From the elements considered to be heavy metals, Hg was found to be significantly increased in patients with PEX compared to controls. In this study, the levels of Zn, Cu, Se, Cr, Ni, and As were not significantly different in the PEX patients compared to healthy controls. The above-mentioned study by Lin et al. also investigated the association between blood levels of some heavy metals, mercury, cadmium, lead, and arsenic, in 2680 South Korean individuals [124]. In this study blood mercury level was positively associated with glaucoma diagnosis, thus confirming its negative impact on human health. Besides, these findings suggest that a higher blood mercury level is associated with an increased incidence of glaucoma. No definitive association was identified for blood levels of cadmium or lead, or urine arsenic level. The association between levels of three heavy metals and the occurrence of open-angle glaucoma (OAG) with low- and high intraocular pressure was studied by Lee et al. (in South Korean teenagers [43]. The mean of blood cadmium levels was significantly higher in subjects with OAG than that of the non-glaucomatous group. In contrast, there were no significant differences in blood lead and mercury. The results of this study suggest that cadmium toxicity could also play a role in glaucoma pathogenesis, particularly in male OAG patients with low baseline intraocular pressure.

Yuki et al. showed that lead accumulation might be an unrecognized risk factor of non-pressure-dependent glaucomatous optic neuropathy [126]. They showed some lead accumulation in Japanese open-angle glaucoma patients. A higher hair lead level, which reflects the total body burden, was associated with POAG in females, especially those with low-tension glaucoma. Lead is known to cause tissue damage by oxidative stress, lipid peroxidation, and DNA damage [127]. Nevertheless, the precise role of lead and other heavy and toxic metals on the appearance of glaucoma remains unknown.

Although the results of these studies indicate the association of elevated levels of some heavy metals with the occurrence of glaucoma, we do not know whether this is due to their general toxicity or whether they have a particular effect on the glaucoma pathogenesis (Figure 2).

## 3. Role of Endothelins

Endothelin-1 (ET-1) is a potent vasoconstrictor peptide released from endothelial cells. ET-1 signals through two G protein-coupled receptors A and B. Both receptors induce an increase in intracellular Ca^2+^, mainly from the extracellular space through voltage-independent mechanisms, the receptor-operated channels, and store-operated channels [27]. Wu et al. showed in in vitro experiments the inhibitory effects of various antiglaucoma drugs on endothelin-1 (ET-1), and KCl induced increase of intracellular free Ca^2+^ in cultured rat vascular smooth muscle cells A7r5 [128]. Ca^2+^ concentration and mobility were analyzed with the fluorescent calcium indicator—fura-2-AM. These studies showed that unoprostone and beta-adrenergic blocking agents (betaxolol, timolol, levobunolol, and carteolol) inhibited ET-1-induced increase of Ca^2+^, according to the mechanism based on blocking extracellular calcium influx via L-type voltage-dependent Ca^2+^ channel in vascular muscle cells.

Endothelins (ETs) are 21-amino acid peptides having a key role in vascular homeostasis [129]. Their actions constrict blood vessels and raise blood pressure, acting through activation of ET receptors, which results in elevation of free intracellular calcium. Their effects are precisely regulated through inhibition or stimulation of ET-1 release from endothelium. ET and ET receptors are considered therapeutic targets for disorders resulting from elevated ET-1 levels, e.g., using ET receptor agonists. In eye diseases associated with increased intraocular pressure, such as glaucoma, ET-agonists are also considered potential therapeutics.

Endothelins represent a vasoconstrictor agent with a possible role in glaucoma and other eye diseases [57]. Increased plasma ET-1 levels have been found in normal-tension glaucoma patients. Furthermore, administration of ET-1 produced an optic neuropathy similar to glaucoma. In the rabbit model of glaucoma, administration of endothelin-1 to the anterior optic nerve region induced a significant decrease in local blood flow [130].

The expression and secretion of ETs are regulated by multiple mechanisms, such as changes in osmolality, an increase in the activity of the renin-angiotensin system, and hypoxia [131]. Most of the effects of ET-1 are associated with an increase in intracellular calcium concentration [68]. Endothelin also inhibits the active Na^+^/K^+^ transport in various tissues. Responses regulated by ET-1 are associated with increases in Ca^2+^ concentration, either by an influx of calcium or the release of intracellular calcium stores. On the other hand, ET-1 may be effectively inhibited by calcium channel blockade. The Ca^2+^ concentration is regulated by ion channels and exchangers (NCX) that control influx, efflux, sequestration, and calcium release. The activation of nuclear endothelin type B receptor by ET1 results in an increase in nuclear calcium by opening R-type calcium channels and Na^+^/Ca^2+^ exchangers on the nuclear membrane [132]. As shown by Okafor and Delamer, ET-1 causes inhibition of active Na^+^/K^+^ transport in the intact porcine lens by a mechanism that involves activation of ET receptors [133]. Activation of ET receptors also causes an increase of cytoplasmic calcium concentration in cultured lens epithelial cells.

A review of the medical literature by Shoshani et al. demonstrated the importance of ET-1 in glaucoma patients [62]. The ET-1 concentration in the plasma was shown to be markedly increased in open-angle glaucoma patients, suggesting that vascular dysfunction and pathology may play a role in pathogenesis. Additionally, aqueous humor ET-1 levels were found to be elevated in high tension glaucoma [13,39]. Increased ET-1 not only reduces optic nerve blood flow and interferes with axoplasmic transport, but also activates astrocytes [13,35]. Once activated, astrocytes upregulate the production of free radicals, NOS and ET-1, thereby creating a microenvironment leading to axonal damage [13,134]. This indicates the potential role of ET-1 antagonists in the therapy of glaucoma. Magnesium indirectly inhibits the activation of astrocytes and prevents neuronal loss via the inhibition of the endothelin pathway [13,135]. Thereby, with the reduction of oxidative stress and vasorelaxation, the role of Mg should be further investigated in the prevention and treatment of glaucoma and glaucoma-related neuronal loss.

The above-cited studies support the view of the crucial role of endothelin, ET receptors, and their cooperation with the Mg^2+^, K^+,^ and Na^+^ transport systems in the etiology of glaucoma and make them important targets for glaucoma treatments.

## 4. Genetic and Genomic Studies

While the effects of trace elements are summarized in Figure 2, the importance of genomic involvement must be emphasized. To date, it is well-know how genetic mutations can leads to accumulation of the aforementioned substrate in tissues and organs including the nervous system, and the eye causing progressive organ dysfunction [136,137]. Genetic and genomic research is mainly based on the search for mutations and genetic polymorphisms in genes related to the metabolism, transport, and biological action of various macro and microelements. The expression of these genes in the eye tissues of animals with artificially induced glaucoma symptoms and in people suffering from this disease is also examined. For example, in case of optic nerve drusen (acellular deposits located in the optic nerve head) even if the pathophysiology remains unknown, they may develop as by-products of impaired axonal metabolism in genetically predisposed individuals [138]. Such studies may lead to understanding the molecular mechanisms of the etiology of degenerative eye diseases, and contribute to the detection of drugs that are effective in the treatment of glaucoma.

To understand the mechanisms mediating retinal ganglion cell loss in glaucoma, Farkas et al. compared the expression patterns of iron and/or copper metabolism-related genes—transferrin, ceruloplasmin, and ferritin—in the normal and glaucomatous retina in monkey and human eyes [21]. In monkeys, laser photocoagulation was used to produce unilateral experimental glaucoma. To examine the retinal expression of iron-related gene transcripts and protein immunohistochemistry, in situ hybridization and quantitative reverse transcription polymerase chain reaction (RT-PCR) were used. Increased mRNA and protein levels of the iron-regulating proteins were shown in glaucoma. Comparison of glaucomatous with control monkeys demonstrated increased mRNA expression of transferrin, ceruloplasmin, and ferritin heavy and light chains. As shown by in situ hybridization, expression of transferrin was localized to the retinal inner nuclear layer, and of ferritin to both the inner and outer nuclear layers. Immunohistochemical examination of monkey and human glaucoma also demonstrated increased levels for these irons and/or copper-related proteins. Together, these results support the view of the involvement of iron and copper metabolism and associated antioxidant systems in the pathogenesis of glaucoma.

The Na/K/Cl co-transporters play a critical role in mediating the transport of ions and water across epithelia and regulating the volume of various cells, including trabecular meshwork (TM) cells, in the eye [139,140]. Some investigations showed that the co-transporter function and regulation are altered in the glaucomatous eye. Putney et al. cultured normal and glaucomatous human TM cells derived from trabeculectomy-treated patients [141]. They showed that in the TM cells removed from glaucoma patients, the Na/K/Cl co-transport activity was reduced by about 30% compared with that of normal cells. Western blot analyses showed that expression of the co-transporter protein (NKCC) in glaucomatous TM cells was reduced by 64%. These findings indicate that Na/K/Cl co-transport function and regulation are altered in glaucoma and that the expression of the Na/K/Cl co-transporter protein can be regulated at the post-transcriptional level.

There are two isoforms of the NKCC proteins (NKCC1 and NKCC2), encoded by two different genes—*SLC12A2* and *SLC12A1*. Janssen et al. performed a gene expression analysis (microarray) of the human ciliary body epithelia (CBE) of post mortem eyes [142]. High expression of several POAG-related genes was found. These genes are involved in the regulation of CBE ion transport and aqueous humor (AH) flow, with possible implications for POAG. The authors hypothesized that mutations in these genes might modify aqueous humor production and composition, and that these genes can be possible new targets for AH lowering drugs. As potentially new drugs that might target these channels, genes are mentioned such as those for the Na^+^/K^+^-ATPase, encoded by *ATP1A1* and *ATP1A2*, two sodium channel proteins encoded by *SCN4B* and *SCN8A*, and the potassium channel proteins encoded by *KCNA5* and *KCNQ1*.

The Na^+^/HCO^3−^ co-transporter (NBC1), encoded by the *SLC4A4* gene, controls systemic pH in part by absorbing filtered bicarbonate. Bok et al. determined the expression pattern of sodium-bicarbonate co-transporter protein (Na^+^-coupled HCO^3−^, in humans encoded by the *SLC4A4* gene) in rat eye [143]. They used specific antibodies to detect the NBC1 protein variants by immunoblotting and immunocytochemistry. It has been shown that NBC1 is expressed in the cornea, conjunctiva, lens, ciliary body, and retina, whereas the expression of kNBC1 is restricted to the conjunctiva. For the first time, evidence for extra-renal kNBC1 protein expression was shown. Vibat et al. found two splice variants of human Na/K/Cl co-transporter protein—hNKCC1a and hNKCC1b—encoded by the *SLC12A1* gene [144], and investigated the possibility of differences in the relative expression of a and b variants in normal versus glaucomatous human trabecular meshwork (TM) cells. The authors have assumed that this phenomenon might underlie the aberrant co-transporter function observed in glaucoma. However, they did not find significant differences between normal and glaucomatous cells with respect to the amounts of NKCC1a and NKCC1b mRNA. These results suggest that the basis for the observed change in co-transporter function in glaucomatous TM cells involves posttranscriptional events rather than the alternative splicing of the *SLC12A1* gene transcript.

Homozygous point mutations in electrogenic sodium bicarbonate co-transporter 1 (NBC1), in humans encoded by the *SLC4A4* gene, cause proximal renal tubular acidosis (pRTA), glaucoma, and cataracts, usually accompanied by mental retardation [145]. Dinour et al. have identified and functionally characterized a novel [146], homozygous, missense mutation (S427L; Ser/Leu substitution) in NBC1, also resulting in pRTA and similar eye defects but without mental retardation [146]. They expressed wild-type NBC1 and mutated S427L-NBC1 in *Xenopus* oocytes and evaluated the Na^+^/HCO^3−^ co-transport function by measuring intracellular pH and membrane currents using microelectrodes. The loss of the function of the mutant transporter was shown. Suzuki et al. identified a homozygous 65-bp deletion (Delta65bp) in the C terminus of NBC1 protein [147], corresponding to the codon change Ser982AspfsX4 in two sisters with pRTA accompanied with ocular abnormalities. Several heterozygous members of this family also presented glaucoma. While transfected to *Xenopus* oocytes, the Delta65bp mutant showed almost no transport activity. These experiments revealed that voltage- and Na-dependent transport by S427L-hkNBC1 is altered, causing both insufficient HCO^3-^ absorption and inappropriate anterior chamber fluid transport, which in humans could be the cause of glaucoma. Demirci et al. identified a novel, homozygous, missense *SLC4A4* mutation (Leu522Pro in kNBC1) in a patient who had pRTA and bilateral ocular disease (cataract, glaucoma, and band keratopathy) [67]. The mutant RNA failed to induce electrogenic transport activity. Knowing these mutations and their effects increases our understanding of the structural/functional aspects of the Na^+^ Na^+^/HCO^3−^ co-transporter (NBC1) protein and the molecular basis of the ocular pathologies associated with its defects.

Igarashi et al. have reported missense mutations in the *SLC4A4* gene [145], which significantly reduced the activity of Na^+^/HCO^3-^ co-transporter. One of these mutations, a c.234C>T transition, resulted in the formation of a stop codon in Na^+^/HCO^3−^ co-transporter kNBC1 (gene *SLC4A4*), and was identified in a patient suffering from pRTA and bilateral glaucoma (Igarashi et al.) [148]. There is also a loss of function mutation resulting in a truncated kNBC1 protein that lacks the 1007 amino acids of the carboxyl-terminus. Co-segregation of this mutation with the disease was observed. These studies may help understand the role of Na^+^/HCO^3-^ co-transporter protein (NBC1) and the molecular mechanisms responsible for abnormalities in ocular electrogenic sodium-bicarbonate co-transport in patients with mutations in the *SLC4A4* gene.

As discussed above, endothelins (ETs) and their receptors are promising targets for therapy of ocular diseases resulting from increased pressure and restricted blood flow in the eye. ET receptors are G protein-coupled receptors, and their activation by ET-1 results in an increase in nuclear calcium by opening calcium channels and Na^+^/Ca^2+^ exchangers (NCX) in the nuclear membranes [132]. Endothelin inhibits active Na^+^/K^+^ transport in various tissues. On the other hand, Mg^2+^ improves blood flow by modifying endothelial function via endothelin-1 (ET-1) and endothelial nitric oxide (NO) pathways. Thus, there is a close relationship between the endothelins effect on intraocular blood pressure and cellular flow of Ca^2+^, Mg^2+^, Na^+,^ and K^+^ ions. Therefore, any changes in the expression level of endothelin and its receptors or mutations in their genes may modify or disturb the ionic flows and make the whole system inoperative.

Endothelin B (ETB) receptor expression was estimated in the rat model of glaucoma [149]. Elevation of intraocular pressure increased expression of ETB receptor in the retina, mainly in RGCs, and caused their loss. This effect was significantly attenuated in ETB receptor-deficient (KO) rats. In addition, degenerative changes in the optic nerve were greatly reduced in KO rats. These results showed that the increased expression of ET receptor B might contribute to a decreased survival of RGCs in glaucoma. In the same laboratory, but later, McGrady et al. examined changes in endothelin A (ETA) receptor in the retinas of ocular hypertension model rats using immunohistochemical analysis [150]. After two and four-week elevation of intraocular pressure, increased expression of ETA receptor protein was observed in retinal sections from rat eyes. In vitro studies showed overexpression of ETA receptors in 661W cells and decreases cell viability of primary RGCs. Thus, the above results showed the involvement of both ET receptors A and B in mediating cell death. These findings raise the possibility of the use of dual receptor antagonists for the treatment of glaucomatous neuropathy.

He et al. studied the effects of endothelins on gene expression profile in primary RGCs both on the mRNA (Real-time PCR) and protein (immunocytochemistry and immunoblotting) levels [151]. Treatment with ET-1, ET-2, or ET-3 resulted in upregulation of 328, 378, or 372 genes, respectively. This would correspond to altered protein expression of proteins having regulatory roles in apoptosis, calcium homeostasis, cell signaling, and matrix remodeling being altered by treatment with endothelins. These results may contribute to the understanding of the molecular mechanisms underlying the contribution of endothelins to RGC death/survival in neurodegenerative changes during ocular hypertension.

Mi et al. used transgenic mice with overexpression of ET-1 in vascular endothelial cells (TET-1 mice) to examine the long-term effect of increased vascular ET-1 on the retinal tissue [152]. These mice carried a transgene that included ET cDNA with SV40 polyA under tyrosine kinase with immunoglobulin and epidermal growth factor homology domain (Tie-1) promoter. The TET-1 mice exhibited a significant progressive loss of RGCs and decreased retinal thickness as early as 10–12 months of age. At 24 months, the retinal degeneration became more severe, with around 30% RGCs loss associated with thinning of the retinal nerve fiber. The IOP level was normal in all ages. These results suggested that TET-1 mice may be a valuable model in addressing endothelial ET-1-related mechanisms in vascular-associated retinal degenerative diseases.

Ishikawa et al. investigated associations between polymorphisms of endothelin (ET-1) and endothelin receptors (ER) A and B genes with the occurrence of glaucoma in Japanese patients [30]. Maximum intraocular pressure and visual field defects were examined. Eight single nucleotide polymorphisms (SNPs) and one “insertion-deletion” (InDel) polymorphism were detected, located either in sense or antisense DNA strands. For all polymorphisms studied, no significant differences of genotype distributions were shown between open-angle glaucoma patients and healthy controls, although for the ET-1 polymorphism 198K>N (base change G/T in exon 5; change of lysine to asparagine) the Lys-Lys (GG) genotype tended to be more frequent than in open-angle glaucoma patients. For the endothelin receptor A gene (*ERA*), the +70C>G polymorphism, located in exon 8, was shown to be associated with visual field defects in normotensive glaucoma patients, the GG genotype being associated with worse visual field defects in NTG patients. Therefore, this polymorphism may be related to glaucoma risk factors. In a similar investigation, Kim et al. have shown that the polymorphism c.1222C>T in ERA, located in the mRNA 3′-untranslated region (3′UTR), was significantly associated with the occurrence of glaucoma in Korean patients [153]. Moreover, the TT genotype showed a younger age at the time of diagnosis compared to CC+CT genotypes. The AA genotype for the c.-231G>A *ERA* gene polymorphism (in exon 1) tended to be associated with a lower intraocular pressure than in the GG+GA genotype group. Kosior-Jarecka et al. studied the possible association of four single nucleotide polymorphisms (SNPs) of endothelin (ET) and endothelin receptor type A (*ERA*) genes and the risk for normal and high-tension glaucoma [154]. A significant difference between the glaucoma patients and healthy controls was found regarding the frequency of the *ERA* genotypes c.1222C>T and +70C>G. Moreover, polymorphic variants of endothelin (198K>N; Lys/Asp substitution) and endothelin receptor type A gene (c.1222C>T, +70C>G, 231G>A) affected the ET plasma concentrations. However, no association was shown between the plasma endothelin levels and risk factors for normal-tension glaucoma.

Although the studied polymorphisms were mostly located in the exons of the *ET* or *ERA* genes, and some of them caused amino acid substitution, their functionality was not confirmed. Therefore, the mechanism of their influence on the pathogenesis and pathophysiology of glaucoma is unclear. It is also unknown whether they are related to the regulation of intraocular blood pressure.

In other studies, the association of various mutations in genes involved in the regulation of ion transport through cell membranes, potentially related to the occurrence of glaucoma, was studied. Liu et al. investigated the association of variants in two candidate genes that are important in cerebrospinal fluid production, aquaporin 1 (*AQP1*) and solute carrier family 4 (sodium bicarbonate transporter; *SLC4A10*), with open-angle glaucoma [155]. Altogether, eleven single nucleotide polymorphisms (SNPs) in *AQP1* and *SLC4A10* were genotyped in glaucoma and control subjects, and genotype/allele frequencies were compared between the patients and healthy subjects, finding no statistically significant differences. Thus, no association was found between common sequence variants in the *AQP1* or *SLC4A10* genes and the occurrence of glaucoma.

Single nucleotide polymorphism in the human *SMOC2* gene (encoding the SPARC-related modular calcium-binding protein 2, SMOC2) was investigated by Al-Dabbagh et al. [24]. SMOC2 regulates the expression of extracellular matrix (ECM) proteins and matrix metalloproteinases (MMPs) which are known to play an important role in the pathogenesis of primary glaucoma. Genotyping of SNP rs13208776 (A/G), located within intron 4 of the *SMOC2* gene, revealed a significantly higher frequency of GA genotype than that of the GG in glaucoma patients compared to controls. The primary angle-closure glaucoma (PACG) patients had a significantly higher frequency of GA genotype, thus indicating that the *SMOC2* gene mutation may be a risk factor. However, the mechanism of the influence of this polymorphism on the etiology of glaucoma is unknown. It can, however, be hypothesized that the nucleotide substitution GA in SPARC2 protein modifies the biological function of the Ca receptor proteins in ocular tissues.

Golubnitschaja et al. investigated *MMP* gene expression in circulating leukocytes isolated from normal-tension glaucoma (NTG) patients compared to healthy controls [156]. The genes were sequenced, and individual mRNA pools were quantified using semiquantitative RTPCR. Upregulation of MMP-9 and MT1-MMP (*MMP-14*) gene transcripts was shown in NTG patients by RT-PCR and confirmed in the protein levels by Western-blot analysis, suggesting an increased enzymatic matrix metalloproteinase activity in these patients, and thus confirming the role of extracellular matrix and matrix metalloproteinases, the zinc-activated enzymes, in glaucoma.

To identify genes associated with open-angle glaucoma and its subtypes, Zhou et al. analyzed potentially damaging genetic variants using whole-exome sequencing [157]. The normal-tension glaucoma cases showed enrichment in mutations of genes encoding ion channel transport proteins, including calcium, chloride, and phospholipid transporters involved in plasma membrane homeostasis. This study also showed that mutations in eye development genes were enriched in glaucomatous tissues. They concluded that high tension glaucoma could result from aberrant responses to protein misfolding while the low-tension disease is associated with impaired plasma membrane homeostasis, increasing susceptibility to apoptosis.

## 5. Perspectives

Excessive calcium and iron consumption was suggested to be related to an increased risk of glaucoma [5]. However, in some studies, controversial results were obtained because dietary supplementation with calcium and iron, known as metabolic oxidants, may reduce human susceptibility to glaucoma [83,158]. There may be a threshold level of iron and calcium intake, above which there is an increased risk of the development of glaucoma. It seems, therefore, that the matter of the influence of iron and calcium in the diet on the incidence of glaucoma is still unclear and requires further intensive research on large human populations. Selenium is a microelement, excessive consumption of which could be toxic [25,123]. It was shown that elevated selenium intake might increase the risk of OAG [49]. However, the data on the possible link between excessive iron and selenium intake and the occurrence of glaucoma is still limited. Many reports suggest a potential role for dietary factors which may influence the occurrence and treatment of glaucomas [5,49].

Understanding the role of trace elements and proteins involved in their metabolism and transport in the etiology and progression of various types of glaucoma might accelerate the development of methods for glaucoma prevention and treatment. However, for most of the studied trace elements, the causal relationship with glaucoma is still unknown. It remains unclear whether additional treatment is responsible for the observed minimal response to supplemental therapy. Indeed, most of the reviewed studies are not randomized trials (RCTs) and describe only qualitative and not homogeneous data. Thus, further RCTs with larger sample sizes and more prolonged treatment are warranted to assess efficacy in treating glaucoma damage and its progression.

Genetic and genomic studies also appear to be important in understanding the role of trace elements in glaucoma. Gene transfer techniques and genome editing may lead to the development of gene therapies for this debilitating disease.

## Figures and Tables

**Figure 1 ijms-22-04323-f001:**
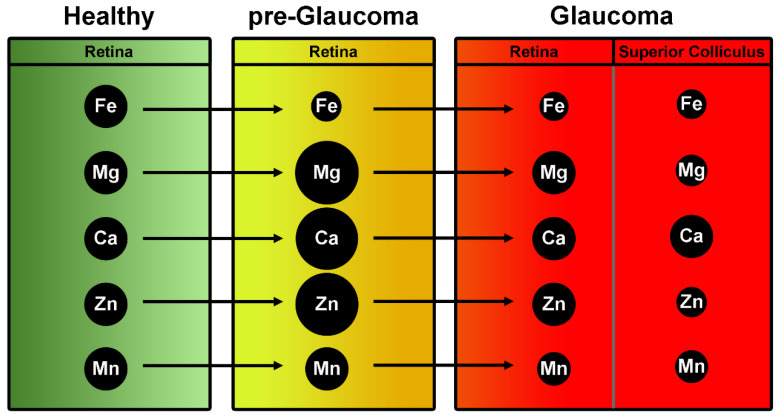
Summary of trace element levels on retinas of a mouse model of glaucoma (DBA2J). Green column indicates the concentration of trace elements expressed in control retinas. In yellow (middle column), levels of trace elements in retinas at pre-glaucomatous stage. In red, trace elements involvement at full-blown stage of pathology. Dimensional change of black spheres simulates the expression of elements at progressive stages of glaucoma.

**Figure 2 ijms-22-04323-f002:**
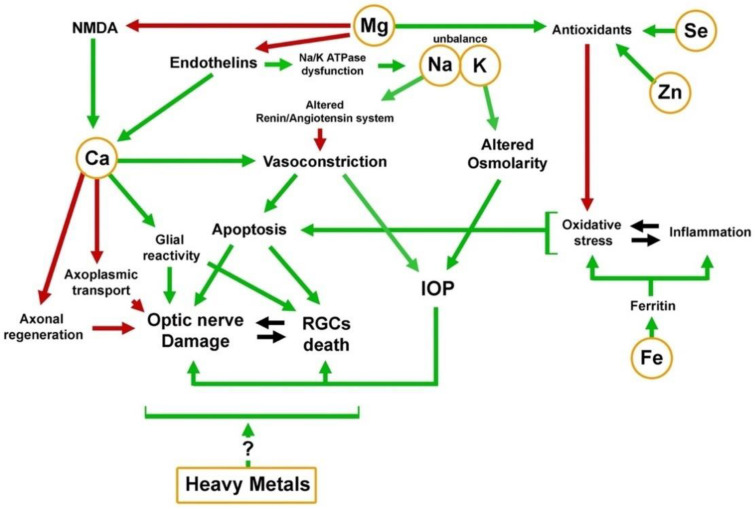
Mechanistic scheme of the activity of elements in the pathological background of glaucoma. Green arrows indicate involvement of the element in activational process; Red arrows indicate inhibition.

**Table 1 ijms-22-04323-t001:** Main studies assessing the role of trace elements in neurodegenerative eye diseases.

Elements	Authors/Year	Main Results
Iron and copper	Farkas R.H., et al., 2014 [21]	Comparison of glaucomatous with control monkey retinas demonstrated increased mRNA expression of iron-regulating proteins. The increased levels of iron-regulating proteins in glaucoma are beneficial, because of their ability to limit iron-related oxidation.
	Gye H.J. et al., 2016 [22]	Serum ferritin has become the preferred marker for assessing iron-related oxidative stress. High serum ferritin levels were independently associated with greater risk for glaucoma.
	Hara H. et al., 1999 [23]	Lomerizine increased cerebral blood flow in animal models, with no significant adverse effects. This suggests that this drug may be clinically useful in conditions associated with circulatory disturbances (such as migraine, normal-tension glaucoma, vertigo and stroke).
Calcium	Al-Dabbagh N. et al., 2017 [24]	Mutation of the SPARC-related modular calcium-binding protein 2 (SMOC2) gene may be a risk factor for glaucoma, probably secondary to modification of the biological function of the Ca receptor proteins in ocular tissues.
	Hartikainen H. et al., 2005 [25]	Calpain, a ubiquitous calcium-sensitive protease, is known to play a role in the neurodegenerative diseases such as cerebral ischemia, Alzheimer’s disease, Parkinson’s disease, Huntington’s disease, multiple sclerosis, and others
	Head K.A. et al., 2001 [26]	Altered calcium homeostasis makes neurons more vulnerable to oxidative stress
	Hohberger B. et al., 2018 [20]	Plasma membrane calcium channel inhibitors were found to arrest acute axonal degeneration and improve regeneration after damage to the optic nerve
	Houde M. et al., 2016 [27]	In vivo studies showed the neuroprotective effects of calcium channel blockers. Moreover, beta-adrenoceptors show calcium channel blocking activity, which may be responsible for the neuroprotective effects.
	Hurst S. et al., 2017 [28]	Animals topically treated with calcium channel blocking agents showed a significant reduction of intraocular pressure in steroid induced glaucoma.
	Iqbal Z. et al., 2002 [29]	Treatment with calcium channel antagonist (nimodipine) significantly improved the visual field and color vision in glaucoma patients.
	Ishikawa K. et al., 2005 [30]	Oral nilvadipine increased the blood flow in distal retrobulbar arteries in normal-tension glaucoma.
	Hara et al., 1999, 2004 [23,31]	The Ca2+ channel blocker lomerizine increases ocular circulation and protects neuronal cells in animal models. It may be useful as a therapeutic drug against retinal diseases that involve a disturbance of the ocular circulation (such as glaucoma and retinal vascular occlusive diseases).
	Kitazawa Y. et al., 1989 [32]	Patients with low-tension glaucoma treated with the Ca2+ antagonist nifedipine with for 6 months. Six patients showed a constant improvement of visual field.
Selenium	Phelps Brown N.A. et al., 1998 [33]	Excessive selenium supplementation may increase the glaucoma incidence
	Prasad A.S. et al., 2014 [34]	High plasma selenium concentration and middle concentration of aqueous humour selenium was significantly associated with glaucoma.
	Prasanna G. et al., 2002 [35]	The mean selenium levels in aqueous humor and in serum of patients with PEX syndrome were lower than in the control group. These results may support the role of impairment in antioxidant defense system in the pathogenesis of PEX syndrome.
	Quigley H.A. et al., 2006 [3]	Selenium supplementation (200 mg/daily) was linked to the development of glaucoma. The risk was even higher in those who continued selenium supplementation after the trial.
Zinc	DeToma A.S. et al., 2014 [36]	Retinas of pre-glaucomatous mice had greater Mg, Ca, and Zn concentrations than those of glaucomatous and greater Mg and Ca than controls
	Grahn B.H. et al., 2001 [37]	Zn supplementation seems beneficial for the patients with diabetes. Zn as an antioxidant attenuates ROS effect, therefore it might protect retina from ROS damage, thereby being protective against DR
	Newsome D.A. et al., 1995 [38]	The concentrations of zinc are reduced in human eyes with signs of age-related macular degeneration (AMD) suggesting that zinc deficiency may lead to oxidative stress and retinal damage
	Noske W. et al., 1997 [39]	Zinc is essential in the in the eye functioning. In retina and retinal pigment epithelium zinc interacts with taurine and vitamin A and modify photoreceptor plasma membranes, regulate the light-rhodopsin reaction, modulate synaptic transmission and serve as an antioxidant.
	Osborne N.N. et al., 2016 [40]	Zinc could be involved in light induced retinal injury; however, the mechanisms of retinal light damage in the pathology of glaucoma remain unknown.
Magnesium	Ekici F. et al., 2014 [13]	Mg may protect retinal ganglion cells from oxidative injury by combined effects on voltage-dependent calcium channels, glutathione synthesis, lipid peroxidation, and maintaining the regulation of many metabolic enzymatic reactions
	Kumar A.R. et al., 2002 [41]	Mg is important for maintaining the structural and functional integrity of several vital ocular tissues
	Laganovska G. et al., 2003 [42]	Mg deficiency has been shown to cause defective neurotransmitter transport mechanism, mitochondrial dysfunction, defective Golgi body function and protein processing dysfunction, neuronal degeneration and apoptosis
	Lee S.H. et al., 2016 [43]	Mg plays a crucial role in Na^+^ and K^+^ transport in cells. It is an important cofactor of Na^+^/K^+^-ATPase
	Lenartowicz M. et al., 2015 [44]	Mg increases blood supply to the optic nerve by dilating the optic blood vessels
	Li X. et al., 2011 [45]	Administration of Mg twice a day for 1 month had beneficial effects on visual field
	Lillico A. et al., 2002 [46]	Treatment with 300 mg of Mg citrate for 1 month did not change the ocular blood flow but caused some improvement in visual field
Molybdenum	DeToma A.S. et al., 2014 [36]	Increased and decreased concentrations of Molybdenum have been observed to affect the illness
Sodium ions/Potassium cations	Jünemann A.G.M. et al., 2018 [47]	Studies on the isolated optic nerve showed that Sodium reduced the influx of sodium and would-be effective neuroprotectants
	Hains B.C. et al., 2005 [48]	Phenytoin (a sodium channel blocker) resulted in neuroprotection of RGCs and optic nerve axons in an experimental animal model of glaucoma
	Ramdas W.D. et al., 2018 [49]	The early studies showed the differences in plasma concentration between normal subjects and glaucoma patients
	Ribas V.T. et al., 2015 [50]	The death of RGCs is a major cause of eye neuropathies. Potassium (K^+^) channels play key roles in modulating the electrical properties of RG cells
	Roy Chowdhury U. et al., 2015 [51]	The presence of Na/K/Cl co-transport activity in trabecular meshwork (TM) cells is able to test the hypothesis that modulation of Na/K co-transport alters intracellular volume and, consequently, permeability of the TM cell membranes.
	Roy Chowdhury U. et al., 2016 [52]	Sodium channel blockade with phenytoin would result in neuroprotection of RGCs
	Roy Chowdhury U. et al., 2017 [53]	Orally delivered phenytoin was effective in protecting neurons, NS-7(4-(4-fluorophenyl)-2-methyl-6-(5-piperidinopentyloxy) pyrimidine hydrochloride a novel Na^+^/Ca^2+^ channel blocker, can protect the rat retina
	Roy Chowdhury U. et al., 2019 [54]	The role of intracellular Na^+^ overload in ischemic injury of acutely isolated rat optic nerves by evaluating electrically potentials (CAPs) from the optic nerves
	Saito S. et al., 2005 [55]	Cromakalim is a hypotensive agent acting via activation of Kir6.2 containing K_ATP_ channels and its effect is additive in combination with the commonly used anti-glaucoma drug latanoprost
	Sakamoto K. et al., 2004 [56]	Derivatives of the K_ATP_ channel were evaluated to control intraocular pressure lowering eye capabilities
	Salvatore S. et al., 2010 [57]	A new class of glaucoma therapeutics, opening the K_ATP_ channels, may have an effect on the trabecular meshwork and intraocular pressure regulation
	Sample P.A. et al., 1986 [58]	K_ATP_ channel openers—diazoxide and nicorandill—lower intraocular pressure by specifically activating the Erk1/2 pathway in ocular cells
	Savigni D.L. et al., 2013 [59]	Application of digoxin, a selective Na^+^/K^+^-ATPase inhibitor, for the α2β3 isoform of the enzyme, efficiently reduces the pharmacologically induced and basal intraocular pressure in rabbits
	Schwalfenberg G.K. et al., 2017 [60]	ATP-dependent potassium channels (K_ATP_ channels) in the mitochondrial or plasma membranes may provide protection against retinal ischemia
	Sheck L., Davies J. et al., 2010 [61]	ATP-dependent potassium channels (K_ATP_ channels) in the mitochondrial or plasma membranes may provide protection against retinal ischemia
	Shoshani Y.Z. et al., 2012 [62]	In Müller glial cells, K_ATP_ channels regulate retinal current and play a key role in retinal protection against ischemic conditions, e.g., ischemic insult
	Siegner S.W. et al., 2000 [63]	MaxiK channels and K_ATP_ channels were found in the eye trabecular meshwork cells
	Silverstone B. et al., 1981 [64]	Opening of potassium channels may play a protective role by increasing the uveal outflow
	Silverstone B.Z. et al., 1990 [65]	KR-31378 ((2S,3S,4R)-N’’-cyano-N(6-amino-3,4dihydro-3-hydroxy-2-methyl-2-dimethoxymethyl-2Hbenzopyran-4-yl)-N’-benzyl-guanidine) as a potent K_ATP_-channel opener, on reducing intraocular pressure and its protective effect on RGCs
	Skatchkov S.N. et al., 2002 [66]	Cromakalim showed a reduction in pressure (by 30–40%) and outflow facility (by 50–80%)
	Sourkes T.L. et al., 1972 [16]	Effects of dexamethasone treatment on Na/K/Cl co-transport activity and co-transporter protein expression in trabecular meshwork (TM) cells
	DeToma A.S. et al., 2014 [36]	Mn concentration was significantly increased in patients with pseudoexfoliation (PEX) syndrome
Manganese	Südhof T.C. et al., 2012 [18]	Mn level was negatively associated with glaucoma diagnosis in a population-based study of South Korean 2680 individuals
	Demirci F.Y. et al., 2006 [67]	Study on the association between levels of three heavy metals and the occurrence of open-angle glaucoma (OAG) with low and high intraocular pressure
Heavy metals [mercury (Hg), lead (Pb), cadmium (Cd), chromium (Cr), nickel (Ni), bismuth (Bi), semi-metals]	Südhof T.C. et al., 2012 [18]	Blood heavy metals level were negatively associated with glaucoma diagnosis in a population-based study on South Korean 2680 individuals
	Tham Y.C. et al., 2014 [8]	The accumulation of Hm may be an unrecognized risk factor of non-pressure-dependent glaucomatous optic neuropathy
	Tykocki N.R. et al., 2010 [68]	Lead is known to cause tissue damage by oxidative stress, lipid peroxidation, and DNA damage

## Data Availability

The raw data supporting the conclusions of this article will be made available by the authors, without undue reservation.

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
