# Peer review of "Influence of Trace Elements on Neurodegenerative Diseases of The Eye—The Glaucoma Model"

_ijms, 2021, doi:10.3390/ijms22094323_

Round 1

Reviewer 1 Report

Congratulations! Nice work and well presented.

Author Response

Dear Reviewer 1,

We are grateful to you for your time and positive comments on our manuscript. Best regards,
All coauthors.

Reviewer 2 Report

  1. While the review itself covers several decades of published investigations and an impressive body of basic and clinical findings up to and including 2018, the organizational structure of this review is difficult to follow and it is unnecessarily verbose. The title suggests a focused review on glaucomatous diseases, but the review is organized by trace elements and their role in numerous neurodegenerative disorders. It would seem to this reviewer that a more coherent concise organizational structure would involve a discussion by glaucoma type (NTG, POAG, exfoliation glaucoma, etc.).
  2. Although well-written, numerous grammatical / stylistic inaccuracies remain throughout this review, requiring extensive editorial assistance.
  3. The use of the terms “micro- and macro-elements” is confusing. What do the authors mean by these terms? Why not just simply use the term “trace elements” to describe these various trace elements?
  4. Many of the statements / paragraphs are reiterations of well-established and well-documented facts that are now commonly found in textbooks. Such common-knowledge statements should be limited to single sentences and appropriately referenced (if needed). I would encourage the authors to focus discussions on recent findings addressing mechanism of action for some of these trace elements as they pertain to specific glaucomatous disorders.
  5. Some statements are superficially vague while others are simply inaccurate (lines 72-73, while iron and copper are well established co-factors needed for neurotransmitter synthesis, zinc and manganese are not directly involved in neurotransmitter synthesis. They are involved in other metabolic processes, but not directly involved in the synthesis of known neurotransmitters.)
  6. Lines 108-109: This statement is inaccurate. As written, it is misleading.
  7. The DBA/2J mouse is best known as a pigmentary dispersion model of pseudoexfoliation glaucoma. (Line 110). It is not a generalized model of glaucoma as inferred.
  8. Figure 1 is not consistent with the text (line 115 states that pre-glaucomatous mice had lower retinal iron concentration, but the figure suggests no difference than control healthy retina). It might be better to show the actual quantified data rather than a qualitative cartoon. Perhaps a table showing the content of trace elements in healthy and glaucomatous types would be more informative.
  9. Line 202: while copper does participate in the Fenton reaction, this reaction is best known to involve reduced iron as the major element that catalyzes the breakdown of hydrogen peroxide forming reactive hydroxyl free radicals. The authors should include this as part of their discussion.
  10. Table 1 might be more informative by adding a column indicating the assessment quantified for each reference cited. As it is now, table 1 organizes the references but is not all that informative.
  11. In many cases throughout this review, the role of trace elements in the pathogenesis of glaucoma is considered equivocal, leaving the reviewer with a “so what” or “probably not relevant” conclusive assessment. A more meaningful discussion might argue in favor of or against a given set of findings per disease type.
  12. Figures 1 & 2 are not referenced in the text. The color distinctions for each are not defined.
  13. Minor grammatical errors: Lines 214, 265, 273, 280.

Author Response

Dear Reviewer 2,

                  We are grateful to you for your time and constructive comments on our manuscript.

We have amended the manuscript according your comments and valuable suggestions. Changes in the last version of the manuscript are reported as red tracked changes.

Below, we also provide a point-by-point response explaining how we have addressed each of your comments.

POINT-BY-POINT RESPONSE

Comments and Suggestions for Authors

  1. While the review itself covers several decades of published investigations and an impressive body of basic and clinical findings up to and including 2018, the organizational structure of this review is difficult to follow and it is unnecessarily verbose. The title suggests a focused review on glaucomatous diseases, but the review is organized by trace elements and their role in numerous neurodegenerative disorders. It would seem to this reviewer that a more coherent concise organizational structure would involve a discussion by glaucoma type (NTG, POAG, exfoliation glaucoma, etc.).

                  Authors’ response:

                  Thanks for the valuable comment. In this article we aimed to summary the current evidence on the role of trace             elements in the pathogenesis and prevention of glaucomatous diseases. Special attention is also paid to the        genetic background associated with glaucoma-related abnormalities of physiological processes that                regulate or involve the ions of elements considered as necessary for the functioning of the cells. To better explain         the connection and the role of trace elements, we have supported the existing evidence underlining the role               shown by some trace elements in different neurodenerative processes. However, our main focus was the role of            trace elements in glaucomatous disorders.

                  To meet the expectations of the reviewer, we have professionally reedited the manuscript deleting all the verbose          parts. We have also re-organized the structure of the manuscript to simplify the entire follow of the manuscript            (please see the manuscript in the tracked-changes modality).

  1. Although well-written, numerous grammatical / stylistic inaccuracies remain throughout this review, requiring extensive editorial assistance.

                  Authors’ response:

            Thanks for the valuable comment. We have edited the grammatical errors as suggested. Additionally, all the                  manuscript has been revised through a professional English editing service (please see the manuscript in the              tracked-changes modality).

  1. The use of the terms “micro- and macro-elements” is confusing. What do the authors mean by these terms? Why not just simply use the term “trace elements” to describe these various trace elements?

                  Authors’ response:

                  Thanks for the comment. As suggested, we have simplified the title of the manuscript, we have also used the term          “trace elements” in the entire manuscript as well.

  1. Many of the statements / paragraphs are reiterations of well-established and well-documented facts that are now commonly found in textbooks. Such common-knowledge statements should be limited to single sentences and appropriately referenced (if needed). I would encourage the authors to focus discussions on recent findings addressing mechanism of action for some of these trace elements as they pertain to specific glaucomatous disorders.

                  Authors’ response:

                  We have substantially edited the general statement and paragraphs and re-organized the entire structure of the              as already reported (please see the manuscript in the tracked-changes modality) The manuscripts was also shortened and carefully edited to focus on the recent specific findings for glaucoma disorders.

  1. Some statements are superficially vague while others are simply inaccurate (lines 72-73, while iron and copper are well established co-factors needed for neurotransmitter synthesis, zinc and manganese are not directly involved in neurotransmitter synthesis. They are involved in other metabolic processes, but not directly involved in the synthesis of known neurotransmitters.).

                  Authors’ response:

Thanks for the comment. We apologize for the misleading sentence. As suggested, we have now rewritten all the sentence.

  1. Lines 108-109: This statement is inaccurate. As written, it is misleading.

                  Authors’ response:

Thanks for the comment. We apologize for the misleading sentence. As suggested, we have now rewritten all the sentence.

  1. The DBA/2J mouse is best known as a pigmentary dispersion model of pseudoexfoliation glaucoma. (Line 110). It is not a generalized model of glaucoma as inferred.

                  Authors’ response:

Thanks for the comment. We apologize for the misleading sentence. As suggested, we have now rewritten all the sentence adding the right details about DBA/2J mouse.

  1. Figure 1 is not consistent with the text (line 115 states that pre-glaucomatous mice had lower retinal iron concentration, but the figure suggests no difference than control healthy retina). It might be better to show the actual quantified data rather than a qualitative cartoon. Perhaps a table showing the content of trace elements in healthy and glaucomatous types would be more informative.

                  Authors’ response:

Thanks for the comment. We apologize for the misleading figure. We have corrected Figure 1. Additionally, a legend explaining the figure has been added to simplify the understanding of the figure as well.

  1. Line 202: while copper does participate in the Fenton reaction, this reaction is best known to involve reduced iron as the major element that catalyzes the breakdown of hydrogen peroxide forming reactive hydroxyl free radicals. The authors should include this as part of their discussion.

                  Authors’ response:

Thanks for the valuable comment. As suggested, we have now included this part regarding the Fenton reaction in from line 257 to line 259.

  1. Table 1 might be more informative by adding a column indicating the assessment quantified for each reference cited. As it is now, table 1 organizes the references but is not all that informative.

                  Authors’ response:

Thanks for the valuable comment. We have improved the Table 1, as suggested.

  1. In many cases throughout this review, the role of trace elements in the pathogenesis of glaucoma is considered equivocal, leaving the reviewer with a “so what” or “probably not relevant” conclusive assessment. A more meaningful discussion might argue in favor of or against a given set of findings per disease type.

                  Authors’ response:

Dear Reviewer, we are grateful for the comment and we agree with it. Unluckely, this review is not a systematic review with metaanalysis and most of included studies are not randomized trials (RCTs). Indeed, most of them just describe qualitative changes and do not show homogeneous data. Therefore, no definitive conclusions can be reached on the base of this literature review.

To date, it remains unclear whether an additional treatment is responsible for the observed minimal response to supplemental therapy. Understanding the role of trace elements and proteins involved in their metabolism and transport in the etiology and progression of various types of glaucoma might accelerate the development of methods for glaucoma prevention and treatment. However, for most of the studied trace elements, the causal relationship with glaucoma is still unknown. It remains unclear whether additional treatment is responsible for the observed minimal response to supplemental therapy. Indeed, most of the reviewed studies are not randomized trials (RCTs) and describe only qualitative and not homogeneous data. Thus, further RCTs with larger sample sizes and more prolonged treatment are warranted to assess efficacy in treating glaucoma damage and its progression.

To meet the reviewer’s wish, we have highlighted these aspects in the Conclusions (please see lines from 1247 to 1261).

  1. Figures 1 & 2 are not referenced in the text. The color distinctions for each are not defined.

                  Authors’ response:

                  Thanks for the valuable comment. We edited Figure 1 as suggested.  Additionally, both figures have been now               referenced in the main manuscript.

  1. Minor grammatical errors: Lines 214, 265, 273, 280.

                  Authors’ response:

            Thanks for the valuable comment. We have edited the grammatical errors as suggested. Additionally, all the                  manuscript has been revised through a professional English editing service.

Looking to receive your favourable considerations

All coauthors

Round 2

Reviewer 2 Report

The authors responses to my stated concerns were well addressed. I have no further comments.